# Coupling Genome-wide Transcriptomics and Developmental Toxicity Profiles in Zebrafish to Characterize Polycyclic Aromatic Hydrocarbon (PAH) Hazard

**DOI:** 10.3390/ijms20102570

**Published:** 2019-05-25

**Authors:** Prarthana Shankar, Mitra C. Geier, Lisa Truong, Ryan S. McClure, Paritosh Pande, Katrina M. Waters, Robert L. Tanguay

**Affiliations:** 1Department of Environmental and Molecular Toxicology, Oregon State University, Corvallis, OR 97331, USA; shankarp@oregonstate.edu (P.S.); mitrageier@gmail.com (M.C.G.); Lisa.Truong@oregonstate.edu (L.T.); 2Biological Sciences Division, Pacific Northwest Laboratory, 902 Battelle Boulevard, P.O. Box 999, Richland, WA 99352, USA; Ryan.Mcclure@pnnl.gov (R.S.M.); paritosh.pande@pnnl.gov (P.P.); Katrina.Waters@pnnl.gov (K.M.W.)

**Keywords:** PAHs, zebrafish, developmental toxicity, morpholino, transcriptomics, *cyp1a*

## Abstract

Polycyclic Aromatic Hydrocarbons (PAHs) are diverse environmental pollutants associated with adverse human health effects. Many studies focus on the carcinogenic effects of a limited number of PAHs and there is an increasing need to understand mechanisms of developmental toxicity of more varied yet environmentally relevant PAHs. A previous study characterized the developmental toxicity of 123 PAHs in zebrafish. Based on phenotypic responses ranging from complete inactivity to acute mortality, we classified these PAHs into eight bins, selected 16 representative PAHs, and exposed developing zebrafish to the concentration of each PAH that induced 80% phenotypic effect. We conducted RNA sequencing at 48 h post fertilization to identify gene expression changes as a result of PAH exposure. Using the Context Likelihood of Relatedness algorithm, we inferred a network that links the PAHs based on coordinated gene responses to PAH exposure. The 16 PAHs formed two broad clusters: Cluster A was transcriptionally more similar to the controls, while Cluster B consisted of PAHs that were generally more developmentally toxic, significantly elevated *cyp1a* transcript levels, and induced Ahr2-dependent Cyp1a protein expression in the skin confirmed by gene-silencing studies. We found that *cyp1a* transcript levels were associated with transcriptomic response, but not with PAH developmental toxicity. While all cluster B PAHs predominantly activated Ahr2, they also each enriched unique pathways like ion transport signaling, which likely points to differing molecular events between the PAHs downstream of Ahr2. Thus, using a systems biology approach, we have begun to evaluate, classify, and define mechanisms of PAH toxicity.

## 1. Introduction

Polycyclic aromatic hydrocarbons (PAHs) are a diverse class of hundreds of compounds, many of which are ubiquitous and persistent in the environment [1,2]. PAHs are detected in a range of environmental matrices including air [3], soil [4], water and sediment samples [5,6,7], and tissues of both animals and plants [6,8]. They originate from both petrogenic sources associated with crude and refined oil products, and pyrogenic sources associated with incomplete combustion of organic materials [9,10]. PAHs enter the environment via both natural (e.g., forest fires and volcanic eruptions) and anthropogenic (e.g., burning fossil fuels, oil refining, and coal tar seal coating) processes [2,11]. Human exposure to PAHs typically occurs through eating food containing PAHs, smoking cigarettes, and inhalation of urban ambient air or wood smoke [2]. Exposure to PAHs has been linked to adverse health effects [12,13,14], with the bulk of research focused on only carcinogenic and mutagenic potential [15,16,17,18].

In 1976, the U.S. EPA placed 16 PAHs on the Priority Pollutant List for regulatory purposes based on the potential for human exposure and carcinogenicity, high frequency of occurrence at hazardous waste sites, and the extent of available toxicological information [19,20,21,22]. The cancer risk for only 27 PAHs including the 16 priority PAHs has been estimated using EPA’s Relative Potency Factor (RPF) approach that benchmarks benzo[*a*]pyrene as the standard to which potency of other PAHs is compared [23]. By benchmarking to a single PAH and a single endpoint, the RPF approach makes potentially inaccurate assumptions about dose additivity and similarity of mechanism of action among PAHs [23,24]. Thus, there is a need to better understand the overall bioactivity of more structurally varied yet environmentally relevant PAH structures to make fully informed toxicity predictions and regulatory decisions.

In addition to known carcinogenic effects, PAHs are implicated in a variety of developmental effects in humans. Epidemiological studies have identified correlations between PAH exposure and decreased head circumference, weight, and length at birth [25], as well as increased childhood obesity and asthma rates [26]. Studies in humans also associate developmental PAH exposure with neurodevelopmental effects, which include increased rates of ADHD, decreased learning and memory capacity [27,28,29], and cardiovascular defects and oxidative stress [30,31,32,33,34,35,36]. While the potential for PAH exposure to cause adverse health effects is clear, the approaches used to study these effects have been insufficient to identify the specific mechanisms of developmental toxicity. Given the diversity of PAH structures, including parent and substituted PAHs such as nitro-, oxy- and methyl-PAHs [37], and the variety of adverse effects they appear to cause, investigators must use more comprehensive approaches to understand the range of hazard potentials and unique mechanisms by which PAHs cause developmental effects. 

Zebrafish (*Danio rerio*) is a vertebrate model used extensively to unravel complex molecular signaling pathways [38,39,40]. Zebrafish biology is favorable for high-throughput chemical testing at a systems biological level [41]. Embryos develop rapidly and externally, are transparent, and are amenable to molecular and genetic techniques [38,42]. The zebrafish genome is fully sequenced, which enables anchoring of complex phenotypes to cellular and molecular events. Zebrafish also possess a remarkably high genetic homology with humans; 76% of human genes have a zebrafish ortholog, and 82% of human genes that cause disease are present in zebrafish, increasing the translational value of the zebrafish model [43]. In zebrafish embryos, developmental exposure to PAHs is associated with reproducible, dose-responsive morphological and behavioral abnormalities [34,44,45,46,47]. The zebrafish model has previously been leveraged to anchor complex exposure phenotypes with their underlying transcriptome changes, enabling understanding of the mechanisms by which PAHs cause developmental toxicity [47]. As a relatively new avenue of research, these studies have only addressed a small fraction of PAH structural diversity [48,49,50]. There is a considerable need to fill the gap in our understanding of structure-mechanism relationships, as they can both reveal therapeutic targets and provide a compelling means of predicting PAH mixture hazard.

Previously, we screened a large library of PAHs in a zebrafish exposure platform [45]. The developmental toxicity profiles of 123 parent and substituted PAHs were characterized in the zebrafish model using a combination of 22 morphological toxicity endpoints, two behavioral assays, and Cyp1a protein expression patterns. The *cyp1a* gene is one of several direct targets of the aryl hydrocarbon receptor (Ahr), is highly induced upon exposure to many PAHs, and is used as a biomarker for activation of the Ahr [51]. Zebrafish have three Ahr isoforms: Ahr1a, Ahr1b, and Ahr2. Only Ahr2 and Ahr1b can bind the potent Ahr ligand 2,3,7,8-tetrachlorodibenzodioxin (TCDD) [52,53], while Ahr1a has been shown to mediate the toxicity of some PAHs [34,54]. Differences in Cyp1a protein expression patterns can suggest binding to the different isoforms [34,47]. PAHs bind to the Ahr isoforms with different ligand-dependent affinities, leading to highly variable developmental toxicity [38,47,52]. Induction of *cyp1a* does not necessarily mediate these developmental toxicity effects [55]. Recent evidence indicates that some PAHs act independently of Ahr activation, further complicating the molecular signaling that leads to toxicity [56]. Thus, there is a need to identify the genes differentially expressed following PAH exposures to help discover PAH targets causally responsible for producing developmental toxicity. Transcriptional profiling gives unbiased insight into both conserved mechanisms and uniquely disrupted transcripts to make sense of how exposure to a given PAH can cause toxicity [49,50]. 

In this study, we sought to characterize, classify, and to begin to define mechanisms of toxicity for PAHs using a combination of developmental toxicity endpoints and genome-wide transcriptomics. In previously published work, the development toxicity profiles of 123 PAHs in zebrafish were found to range from acutely toxic to biologically inactive [45]. Based on the phenotypic effects, we sorted the 123 PAHs into eight bins and selected 16 representative PAHs (one to four PAHs from each bin) to conduct whole-animal transcriptomics at a single concentration, and at 48 h post fertilization (hpf), prior to the onset of visible adverse phenotypes assessed at 120 hpf. To understand the similarities and differences between the transcriptional responses, we used the Context Likelihood of Relatedness (CLR) algorithm that, based on the coordinated responses of genes to exposure, grouped the 16 PAHs into two broad clusters, one of which consisted of the more developmentally toxic PAHs. Overall, we demonstrate the feasibility of using high throughput screening data, transcriptomics, and gene silencing as a systems approach to identify differences in PAH toxicity mechanisms.

## 2. Results and Discussion 

### 2.1. Classification of 123 PAHs into Eight Bins and Characterization of Bins

A previous study assembled and tested a 123 compound library of parent and substituted PAHs for developmental toxicity in embryonic zebrafish [45]. The PAHs were screened across a range of five concentrations between 0.1 and 50 µM for 22 morphological endpoints, two behavioral photomotor response assays (embryonic photomotor response (EPR) at 24 hpf and larval photomotor response (LPR) at 120 hpf), and five tissue localizations of Cyp1a (vasculature, liver, skin, neuromasts, and yolk). PAH developmental toxicity ranged from completely inactive to acute mortality. For this study, we used hierarchical clustering of all phenotypic endpoints from Geier et al. 2018 to classify the 123 PAHs into eight bins of developmental toxicity (Figure 1, Appendix A) for selection of a subset of the PAHs for gene expression analysis. Of the eight bins, Bin 1 had the most phenotypic effects, and Bin 8 PAHs were the least developmentally toxic to zebrafish. Bins 1–3 included PAHs that produced morphological and behavioral effects, Bins 4–7 contained PAHs with only behavioral effects and no morphological effects, and Bin 8 consisted of PAHs that did not cause morphological or behavioral effects (Table 1). PAHs in Bins 1–5 induced either no Cyp1a protein expression or expression in up to four of the five locations identified during the 123 PAH screen [45]. PAHs in Bins 6–8 did not induce Cyp1a protein expression. 

The 123 PAHs included nitrated, oxygenated, hydroxylated, methylated, heterocyclic, and aminated substitutions [45]. Bin 1 (35 PAHs) was the largest and was most associated with high mortality and numerous morphological and behavioral defects (EPR and LPR). This bin was dominated by hydroxylated (11), oxygenated (10), and nitrated (7) PAHs. Bin 2 (14 PAHs) induced fewer morphological defects than Bin 1 and was most associated with LPR behavioral defects. Bin 3 (9 PAHs) PAHs were associated with comparatively fewer morphological defects and no behavioral defects. Bin 4 (17 PAHs) was the second largest with equal representation of methylated (6) and parent (6) PAHs and was most associated with causing LPR defects. Bin 5 (19 PAHs) was dominated by parent (7) and nitrated (5) PAHs that caused no morphological effects, but had behavioral defects (EPR and LPR). Bin 6 (12 PAHs) led to an aberrant LPR in both the lighted (VIS) and dark (IR) phases of the assay, and Bin 7 (10 PAHs) caused an aberrant LPR only in the lighted phase. PAHs in the smallest bin, Bin 8 (7 PAHs), produced no phenotypic effects at the tested concentrations for any of the measured endpoints. The 7 PAHs in Bin 8 were either parent (3), nitrated (2), heterocyclic (1), or methylated (1).

Examination of the bioactivity of substituted PAHs is a relatively recent expansion of the field. In our 123 PAH screen, the most developmentally toxic bin (Bin 1) consisted mostly of substituted PAHs, suggesting that they may be more toxic than parent PAHs. For example, the parent PAH fluoranthene was placed into Bin 5, while its derivatives 3-hydroxyfluoranthene, 3-nitrofluoranthene, and 2-nitrofluoranthene were placed into Bins 1, 2, and 3, respectively [47]. The reliance on benzo[*a*]pyrene as a reference for toxicity estimates in regulatory decisions is also concerning, as it is far less developmentally toxic than many substituted (and parent) PAHs in our screen [45]. Benzo[*a*]pyrene, a parent PAH, clustered into Bin 5 in our screen. Our results demonstrate that substituted PAHs can be more bioactive than their parent compounds and suggest the need to include more substituted PAH classes in hazard modeling and risk assessment. Other recent studies uphold our finding that substituted PAHs are often similarly or more bioactive than the respective parents [57,58,59]. Thus, the current EPA priority PAH list may not only underestimate hazard at contaminated sites, but can also inaccurately inform PAH mixture models.

Only the PAHs within Bins 6–8 had identical developmental toxicity profiles; all PAHs in Bin 6 were associated with an aberrant LPR effect in both the dark and light phases, all PAHs in Bin 7 were associated with an aberrant LPR effect in the light phase, and all PAHs in Bin 8 were biologically inactive in our assays. The PAHs within Bins 1–5 had highly variable developmental toxicity profiles, with no two PAHs resulting in identical profiles. For example, 1-hydroxypyrene and xanthone were two PAHs in Bin 1 that localized Cyp1a expression at 120 hpf in the liver. Exposure to 1-hydroxypyrene caused 24 and 120 hpf mortality, yolk sac edema, and EPR and LPR behavior defects. The lowest effect level (LEL) concentrations for EPR and LPR defects were ≤11.2 µM. On the other hand, xanthone negatively affected many more morphological endpoints, was less behaviorally bioactive, and the LEL was 50 µM for all affected endpoints. Despite these differences in phenotypic effects, the two PAHs were unequivocally placed in the same bin. Cyp1a protein localization was also variable within each bin. For example, in Bin 1, 3-nitrobenzanthrone and 11-H-benzo[b]fluoren-11-one produced several morphological and behavioral malformations, and localized Cyp1a protein at 120 hpf in two locations. However, pyrene-4,5-dione, perinaphthenone, and benzanthrone induced several morphological and behavioral malformations, but did not localize Cyp1a protein at 120 hpf [49]. The differences in responses associated with PAHs within the same bin suggest potentially varied molecular signaling events which were not captured by just the phenotypic screening data presented in Figure 1. It is important to note that the developmental toxicity profile of each PAH was characterized based on nominal water concentration only, and to acknowledge that different uptake rates between PAHs could affect the internal dose and thus the comparisons of outcomes between chemicals. Further, PAHs are known to sorb to polystyrene testing chambers used for these assays which would reduce the available concentration to which the zebrafish is exposed [60,61]. Different chemical properties like hydrophobicity contribute to different rates of adsorption to polystyrene [62], confounding comparisons of uptake in the absence of careful mass balance accounting of chemical fate. For example, a study found that the sorptive losses associated with PAHs with two or three rings was 10% or less compared to PAHs with four or more rings that had sorptive losses of 40 to 70% [62]. Another study demonstrated that there was a higher measured percent sorption to polystyrene at lower exposure concentrations [60]. Thus, various factors likely contribute to the variety of developmental toxicity profiles, and thus the bins the PAHs are grouped into [60]. Future work investigating these potential differentials may provide more accurate determination of potential health impacts for the PAHs presented here.

### 2.2. Selection of 16 Representative PAHs for RNA Sequencing from the Eight PAH Bins

We selected 16 PAHs in total (one to four representatives from each bin) with which to conduct genome-wide transcriptomic profiling at 48 hpf. To make comparisons across the PAHs, zebrafish were exposed to either the EC_80_ concentration of each PAH, or the maximum concentration tested during the phenotypic screening [45] if an EC_80_ concentration was not attainable. For each of the eight bins defined by morphology or mortality effects (Bins 1, 2, 3), candidates were filtered for commercially available standards, an attainable EC_80_, and minimal 24 hpf mortality, then prioritized by known relative environmental abundance [63,64]. For bins defined by behavioral endpoints or no observable developmental toxicity (Bins 4 - 8), candidates were filtered for the availability of larger volume standards and prioritized by known relative environmental abundance. Due to the high variability of the Cyp1a expression patterns within each bin, Cyp1a localization was a secondary filter after the morphological and behavioral endpoints were taken into consideration. After applying the criteria to the bins, one to four representatives from each bin were selected for RNA sequencing, resulting in a total of 16 PAHs representing eight general developmental toxicity profiles: Bin 1: 4h-cyclopenta[def]phenanthrene-4-one (4h-CPdefP) and Retene, Bin 2: Benzo[*k*]fluoranthene (BkF), 3-nitrofluoranthene (3-NF), Benzo[*j*]fluoranthene (BjF), and Carbazole, Bin 3: Dibenzo[*a,i*]pyrene (DB(a,i)P), Bin 4: Dibenzo[*a,h*]pyrene (DB(a,h)P) and 9-methylanthracene (9-MA), Bin 5: Benzo[b]fluoranthene (BbF) and Fluoranthene, Bin 6: 1,5-dimenthylnaphthalene (1,5-DMN), Acenaphthene, and 2-methylnaphthalene (2-MN), Bin 7: Phenanthrene, and Bin 8: Anthracene (Table 2 in methods). 

Transcriptomics provides mechanistic information and identifies biomarkers that are both common and unique to different chemicals. Only a few studies to date have investigated transcriptomic responses to PAH exposure during development [49,50]. Consequently, we have very little mechanistic understanding of how exposure to PAHs causes developmental toxicity. We previously measured a large battery of phenotypic endpoints using our zebrafish high-throughput screening assay [45], and while developmental toxicity profiles were highly variable, many of the individual endpoints were common across the 123 PAHs. High correlation between different endpoints, for example pericardial edema and bent body axis, also limits our ability to infer mechanistic information from the phenotypic data alone [65]. Querying the transcriptome offers a deeper understanding of how structurally distinct PAHs cause different, but often overlapping developmental toxicity profiles. PAH exposure was between 6 and 120 hpf, which is throughout most of zebrafish development. Thus, any perturbations to the complex signaling occurring in that period would be expected to alter some aspect of development [41]. Here, we pursued the next logical dimension, comprehensive transcriptomic analysis using RNA sequencing to classify structure-bioactivity relationships and to identify signaling pathway similarities and differences between 16 PAHs. Transcriptomics at 48 hpf in zebrafish is a desirable time point because it is likely to be close to the molecular initiating event(s) of a chemical exposure, and it precedes most observable toxicity, offering gene expression changes that may be predictive [49,66]. To our knowledge, with the exception of retene, whole-animal developmental transcriptomic evaluations of the PAHs in this study have not previously been investigated [67]. 

### 2.3. Number of Differentially Expressed Genes (DEGs) Generally Correlates with PAH Developmental Toxicity Profiles

Differentially expressed genes (DEGs) were defined as having a fold-change (FC) compared to the control of at least 1.5 and adjusted p-value (PADJ) <0.05. Based on these criteria, 3-NF and 1,5-DMN had no associated DEGs (Figure 2). For each PAH, a full list of DEGs with their associated FC and PADJ are provided in Appendix A. 

Six of the 16 PAHs produced modest transcriptional responses less than or equal to 10 DEGs. The transcriptional responses mostly agreed with their developmental toxicity profiles: five of these 6 PAHs were in the less developmentally toxic bins (Bins 6, 7, 8). 1,5-DMN, acenaphthene, 2-MN, and phenanthrene produced only LPR effects, while anthracene did not cause morphological or behavioral effects. These PAHs were also characterized by a lack of Cyp1a protein expression. The only exception was 3-NF (Bin 2), which had no associated DEGs, but 3-NF exposure produced aberrant circulation. The DEGs associated with acenaphthene, 2-MN, phenanthrene, and anthracene are potential biomarkers of their respective exposures. The remaining 10 PAHs that produced different combinations of morphological and behavioral effects, and Cyp1a protein expression patterns, resulted in varying numbers of DEGs from 21 to 236 (Figure 2). DB(a,h)P from Bin 4 which produced aberrant behavioral effects and Cyp1a localization in the vasculature, liver, skin, and neuromasts, was associated with the greatest number of DEGs at 236. The two PAHs from the most developmental toxic Bin 1, 4h-CPdefP and retene, were associated with 51 and 89 DEGs, respectively. We note that Cyp1a protein expression patterns did not always correlate well with DEGs, even for highly related structures. For example, 4h-CPdefP and retene were both associated with DEGs, but while 4h-CPdefP did not induce Cyp1a protein expression, retene produced vasculature Cyp1a protein expression at 120 hpf. Likewise, DB(a,h)P and DB(a,i)P exposures localized Cyp1a to the same 4 locations, but were associated with 236 and 44 DEGs, respectively. 

The PAHs in Bins 6–8 which had little to no developmental toxicity also effected ≤10 DEGs. To determine if low uptake was the explanation for a lack of phenotypic effects and low number of DEGs, we mined a previously published dataset that quantified the body burden of 48 hpf embryos for 6 of the 16 PAHs (Appendix A) [68]. The 6 PAHs were retene (Bin 1), BbF and fluoranthene (Bin 5), acenaphthene and 2-MN (Bin 6), and phenanthrene (Bin 7). Briefly, the body burdens (nmol/embryo) of the PAHs were measured at three concentrations, 5.39, 11.6, and 25 µM [68]. Concentration uptake ratio (CUR) was defined as the ratio of nominal PAH concentration in the exposure medium to measured concentration in the embryo. For each PAH, the mean and standard deviation of nine CUR values for different nominal medium concentrations (three concentrations) and replicates (three for each concentration value) were calculated and then log-transformed. For fluoranthene, acenaphthene, 2-MN, and phenanthrene, we found that the CUR positively correlated (r^2^ = 0.9, *p*-value = 0.03) with the number of DEGs (Appendix A). The four PAHs had the same nominal exposure concentration (50 µM), and log K_OW_ <5.5. The positive correlation diminished when retene and BbF, two PAHs with log K_OW_ >6 were included in the analysis. The uptake CURs were within the range of the other four PAHs, but the number of DEGs associated with exposure to retene and BbF was 4–100 times higher than those associated with the other four PAHs. These findings demonstrate that the number of DEGs do not broadly correlate with the CUR, at least when PAHs with log K_OW_ >5.5 are considered. More importantly, they suggest that the minimal or lack of developmental toxicity for PAHs like fluoranthene, acenaphthene, 2-MN and phenanthrene, may not be because of a lack of uptake into the zebrafish. We note that we defined DEGs as those genes with at least a 1.5-fold change in expression compared to the controls to identify key genes involved in bioactivity. It is possible that certain genes are associated with developmental phenotypic effects but not captured by our criteria. Therefore, although the number of DEGs can give a very high-level overview of transcriptomic response, it is only an approximation of the true PAH transcriptomic profiles.

### 2.4. Cluster Analysis of Response to PAHs Reveals Two Broad Clusters

In order to identify PAHs with similar and unique transcriptomic responses, we performed hierarchical clustering of PAHs using expression profiles for the 500 genes with the highest coefficient of variation (CV) across the 18 conditions (16 PAH treatments and 2 controls) (Figure 3A). For a list of the top 5000 genes with the highest CV across the 18 treatments, see Appendix A. PAHs clustered into two broad groups. With the Context Likelihood of Relatedness (CLR) algorithm, we inferred a network that links the PAHs based on the coordinated transcription of genes as they respond to those PAHs. We again used only the top 500 genes based on the CV to ensure a more relevant network as basal level responses of genes with a low CV are not present to dilute the effect of the most responsive genes. This analysis also clustered the 16 PAHs into the same two broad clusters (Figure 3B).

The larger cluster (Cluster A) (Figure 3A,B) showed more varied Cyp1a localization patterns and modest or no phenotypic response to PAH exposure. Many of these PAHs, specifically 1,5-DMN, 2-MN, 3-NF, acenaphthene, anthracene, and phenanthrene also showed zero or very few DEGs. All of these PAHs except 3-NF are from the less developmentally toxic Bins 6–8, and share connectors with the control samples (Figure 3B). This indicates the similarity of the transcriptomic responses of these PAHs to the control samples at 48 hpf. A strong overlap between phenotypic effects and transcriptomic changes was seen with some PAH pairs including acenaphthene and phenanthrene, and to a lesser degree, 2-MN and anthracene. The Cluster A PAHs that do not share connectors with either of the control samples are 4h-CPdefP (Bin 1), 9-MA (Bin 4), and fluoranthene (Bin 5). We note that these three PAHs had both varied developmental toxicity responses and no DEGs in common (Appendix A). The DEGs that are unique to each of these PAHs are biomarkers of their respective exposures. 

The smaller cluster (Cluster B) (Figure 3A,B) consisted of retene, BkF, BjF, DB(a,i)P, DB(a,i)P, and BbF, all PAHs belonging to the more developmentally toxic bins 1–5. These six PAHs localized Cyp1a protein expression in the vasculature. All of them except retene localized Cyp1a protein expression in the skin at 72 hpf and 120 hpf, unlike the Cluster A PAHs which did not localize Cyp1a protein in the skin. There were other similarities between Cluster B PAHs: BjF and BkF (Bin 2) both showed an unusual caudal fin phenotype [45], and are structurally similar five-ring PAHs with only one of the rings positioned differently [69]. These PAHs are also adjacent to each other in our transcriptomic response dendrogram (Figure 3A), showing that transcriptomic and phenotypic responses of these PAHs are tightly correlated. The other two PAHs in Bin 2 (carbazole and 3-NF) are less structurally similar and did not cluster tightly with each other or with BkF and BjF, further suggesting different molecular signaling events. DB(a,h)P (Bin 4) and BbF (Bin 5) had the most similar gene expression changes depicted by their close proximity in the dendrogram (Figure 3A), and the thickness of the connector between the two PAHs (Figure 3B). Both PAHs despite being in different developmental toxicity bins (Figure 1), produced aberrant behavioral phenotypes with no malformations. 

Interestingly, there were PAH pairs with similar developmental toxicity profiles that binned together in our phenotypic screening heatmap (Figure 2) yet had less related transcriptomic responses (and vice versa). Retene and 4h-CPdefP are highly disparate in the dendrogram in Figure 3A and clustered separately in our network analysis (Figure 3B) but had similar developmental toxicity profiles (high mortality with morphological and behavioral endpoints in common). The same was observed for the Bin 4 PAHs DB(a,h)P and 9-MA which caused behavioral effects, but were far apart in the dendrogram and the network (Figure 3A,B). Cyp1a expression patterns were also different between these pairs of PAHs. Both the differences in Cyp1a expression pattern and the lack of clustering based on transcriptomics strongly suggest unique signaling mechanisms between these chemical pairs. The fact that PAHs, and many other classes of chemicals, can manifest the same developmental toxicity endpoints in a vertebrate model [70,71], but have vastly different molecular signaling events underlying those endpoints, has often been regarded as a limitation of the zebrafish model, i.e., the responses are not specific enough. The utility of embryonic zebrafish is as a systems level biosensor of chemical insult. Development is the most sensitive period because it represents a highly coordinated and regulated framework of transcription and signaling events. Any perturbation is likely to manifest as abnormal morphology or behavior [41], therefore providing an efficient and sensitive hazard detection model. Our results highlight the importance of coupling developmental phenotypic effects with comprehensive transcriptomic data to understand the mechanisms by which chemicals cause toxicity. 

Clustering of the PAHs in this way reveals how PAHs group together based on similarity in transcriptomic responses. As we conduct more RNA sequencing studies, we can examine the proximity of each new PAH to other PAHs in the network, and expand our classification of PAHs based on gene expression profiles. A potential limitation of our study may be that we chose only a single time point (48 hpf) for comparative RNA sequencing. If a PAH’s bioactivity was entirely dependent on its metabolites, its developmental window of bioactivity may not begin until after 24 hpf, which may explain the lack of developmental phenotypes at this time point in our phenotypic screen. If metabolism of the PAH is most effective in the liver, the major site of metabolism for many chemicals, then the chemicals may not be converted to their metabolites until 72 hpf when the liver is fully developed [72]. By sampling at 48 hpf, we may have preceded the relevant gene expression changes. Similarly, if the PAH was bioactive immediately after the beginning of the exposure, our 48 hpf transcriptomic data may no longer reflect the phenotype’s causative events. A previous microarray study of transcriptional responses found that three PAHs, benz(a)anthracene, pyrene, and dibenzothiophene, differentially regulated mRNA expression between the 24 and 48 hpf time points, highlighting the importance of a time-dependent transcriptional response study [50]. Our single time point/single concentration approach emphasized breadth over depth in querying PAH transcriptome changes. It enabled us to examine all 16 PAHs in our study set, whereas only a small fraction of the structural diversity could have been investigated with multiple time points and concentrations. Future work conducting transcriptomic network analyses like the CLR program presented in this study, for multiple concentrations of a PAH at different time points of zebrafish development will be beneficial to gain a comprehensive understanding of its bioactivity.

### 2.5. Cyp1a Expression is An Inadequate Biomarker for PAH Developmental Toxicity, but is Associated with Transcriptomic Response

To assess Ahr activation, we investigated the correlation between the developmental toxicity profile and transcriptomic responses, and *cyp1a* transcript levels across the 16 PAHs. A widely used biomarker of Ahr activation is *cyp1a* because of its sensitive and robust induction upon exposure to many PAHs [55,73]. *Cyp1a* is also primarily involved in the metabolism of xenobiotics. In general, there was no correlation between developmental toxicity bin number and *cyp1a* expression (Table 3). For example, the most developmentally toxic PAHs were 4h-CPdefP and retene (Bin 1), but *cyp1a* was significantly elevated only in the retene exposure. The low *cyp1a* log_2_FC value associated with 4h-CPdefP exposure, in combination with the lack of Cyp1a protein localization at 120 hpf, suggests a non-Ahr dependent mechanism of action for 4h-CPdefP, an oxygenated PAH which may not require metabolic activation to manifest toxicity [74]. Amongst the 123 PAHs in Geier et al., 2018, there were 18 oxy-PAHs, three of which induced expression of Cyp1a protein in the vasculature, one in both the vasculature and the liver, one in both the liver and the yolk, and one in only the yolk. The remaining 12 did not induce Cyp1a expression at 120 hpf [45]. We note that previous studies have shown that oxy-PAHs can elicit their biological effects through the Ahr [34,58], however our data suggest that 4h-CPdefP acted independently of Ahr.

The Cluster A PAHs, carbazole (Bin 2), 3-NF (Bin 2), 9-MA (Bin 4), and fluoranthene (Bin 5) had lower *cyp1a* expression levels relative to the other PAHs in their respective bins. Despite the low induction of *cyp1a*, exposure to these PAHs resulted in phenotypic responses. The low *cyp1a* expression levels at 48 hpf suggest either a different isoform of the Ahr was activated, or that the 48 hpf time point is not when *cyp1a* is maximally induced for these PAHs [34]. Exposure to carbazole, 9-MA, and fluoranthene induced Cyp1a protein expression in the liver and faint vasculature expression (data no shown). This suggests a possible dominant role for Ahr1a, which was not explored in this study. Previous work demonstrated that Ahr1a mediates Cyp1a protein expression in the liver induced by xanthone, leflunomide, and pyrene [34,54,55]. Cyp1a yolk expression induced by 3-NF and 24 PAHs in the 123 PAH screen is still unexplored in zebrafish [45]. However, the expression of an isoform, *cyp11a1,* was previously detected in the yolk syncytial layer [75,76]. Activation of different Ahr isoforms by the PAHs could be the reason for both differential transcriptomic clustering of PAHs from the same developmental toxicity bin, and for different *cyp1a* induction levels. The remaining PAHs (Bins 6–8) were not associated with morphological malformations or Cyp1a protein expression (Table 1 and Figure 2) and did not have significant *cyp1a* expression levels. These PAHs were associated with either behavioral effects (2-MN, acenaphthene, 1,5-DMN, phenanthrene) or had no observable effects (anthracene), with only anthracene having a positive correlation between the absence of phenotypic effects and low *cyp1a* expression. 

Unlike developmental toxicity bin number, we detected a correlation between PAH transcriptomic response and *cyp1a* transcript levels (Table 3). No Cluster A PAH had significant *cyp1a* transcript levels (*cyp1a* log_2_FC <1, PADJ >0.05). Conversely, *cyp1a* was significantly elevated in all Cluster B PAH exposures at 48 hpf (*cyp1a* log_2_FC >1, PADJ <0.05). Our results suggest that the induction of *cyp1a*, in combination with the Cyp1a protein expression patterns, is an early biomarker of general xenobiotic Ahr activation and downstream transcriptomic changes, but an inadequate biomarker of potential for morphological and behavioral effects of PAHs. 

### 2.6. Cluster B PAHs Activated Ahr2

To evaluate Ahr2 activation upon exposure to each of the six Cluster B PAHs, transient knockdown of Ahr2 via a translational blocking antisense morpholino was performed prior to exposure. Spatial Cyp1a protein expression in 72 hpf control and Ahr2 morphant samples exposed to each of the six PAHs was detected using immunohistochemistry (IHC). Control morphant samples collected at 72 hpf expressed Cyp1a in the same locations as previously identified at 120 hpf (Figure 2): retene: vasculature; BjF and BkF: vasculature, skin, neuromasts; DB(a,i)P and DB(a,h): vasculature, skin, neuromasts, liver; BbF: vasculature, skin (Figure 4). DMSO control fish had low background signal relative to the PAH treatments. In general, the absence of Ahr2 during development severely decreased Cyp1a protein expression associated with exposure to all 6 PAHs (Figure 4). Specifically, expression in the skin induced by five of the six PAHs (all except retene) was completely Ahr2-dependent. When Ahr2 was knocked down, there was a partial reduction of vasculature expression associated with all six PAHs. Cyp1a expression in the liver as a result of exposure to DB(a,h)P and DB(a,i)P was not reduced by Ahr2 knockdown, which suggests an Ahr1a role in Cyp1a liver expression that is consistent with previous studies [34,54,55]. The complete inhibition of Cyp1a expression in the neuromasts upon DB(a,h)P and DB(a,i)P exposure with Ahr2 knockdown indicates that Ahr2 is the primary Ahr isoform responsible for Cyp1a expression in the neuromasts, also consistent with previous work [47].

One commonality among the Cluster B PAHs was the Cyp1a expression in the vasculature at 72 and 120 hpf. Previous studies have shown that induction of Cyp1a protein expression in the vasculature is indicative of Ahr2-dependent toxicity [47,49]. All of the PAHs in Clusters A and B that induced any visible Cyp1a protein expression induced Cyp1a in the vasculature. Only the PAHs in Cluster B induced Cyp1a protein expression in the skin at 72 and 120 hpf, with the exception of retene, which induced Cyp1a protein expression only in the vasculature at these time points. For all five of these PAHs, Ahr2 knockdown led to a complete loss of Cyp1a protein expression in the skin compared to only a partial reduction of the vasculature expression. Previously, of the five PAHs, only exposure to BkF was associated with Cyp1a protein expression in the skin or epidermal cells [77]. We investigated Cyp1a protein expression upon exposure to retene at both 24 and 48 hpf. In addition to the expected vasculature expression, Cyp1a expression was observed in the skin at both time points (data not shown). By 72 hpf, the skin expression shifted to the vasculature. These data suggest that Cyp1a protein expression in the skin may have broader temporal use as a biomarker of Ahr2-activation, reporting as early as 24 hpf, and Ahr2 knockdown leading to its complete loss. We limited our investigation to 72 hpf to detect Cyp1a protein localization in the liver, which is not fully developed and visible until 72 hpf [72].

A definitive test for Ahr2-dependent toxicity would assay morphological and behavioral effects in the complete absence of Ahr2. Because permanently knocking out Ahr2 causes background behavioral effects, in addition to low fecundity [78], we relied on transient knockdown [79,80,81] with the caveat that time points were restricted to no later than 72 hpf. Of the six Cluster B PAHs, only retene, BjF, and BkF caused morphological malformations. A previous study identified that the cardiac effects of retene were Ahr2-dependent [82]. The toxicity of BkF is more complicated; a previous study showed that BkF associated cardiotoxicity was not affected by Ahr2 knockdown [77]. We found that the caudal fin malformation associated with both BkF and BjF failed to develop in the absence of Ahr2 [83]. The Ahr2-dependent caudal fin toxicity, in combination with the almost complete loss of vasculature, skin, and neuromast Cyp1a protein expression in the absence of Ahr2 suggests that BkF and BjF act primarily through Ahr2. BbF, another five-ring PAH that did not cluster with BkF and BjF based on developmental toxicity phenotypes (Figure 2), was part of transcriptional Cluster B, and induced Ahr2-dependent Cyp1a expression in the skin. The differences in phenotypic effects between BbF, BkF, and BjF despite Ahr2 activation suggest one or more of the following: a potential role of the other Ahr orthologs in BbF’s bioactivity, differential PAH stability in solution or uptake into the embryos, and thus dose differences in the embryos, or metabolism of these parent PAHs to metabolites with varying bioactivity. With the exception of DB(a.h)P, the remaining PAHs only caused LPR effects at 120 hpf, precluding a transient knockdown test of Ahr ortholog dependency. Cyp1a protein expression at 72 hpf was instead used as a biomarker for Ahr2 activation. We recognize that the reduction of Cyp1a protein expression upon Ahr2 knockdown is not an indication of Ahr2-dependent toxicity for these PAHs, as Cyp1a expression does not seem to be indicative of morphological or behavioral malformations. Rather, it suggests that these PAHs primarily bind and activate Ahr2.

### 2.7. Cluster B PAHs that Activate Ahr2 Uniquely Enrich Several Pathways 

After identifying the dominant role of Ahr2 in the bioactivity of Cluster B PAHs, we examined the genes and pathways that were enriched to gain insight into the mechanisms of their toxicity. We identified the DEGs uniquely altered by each of the 6 PAHs (Figure 5): Retene—45, BjF—10, BkF—23, DB(a,i)P—8, DB(a,h)P—123, BbF—11. A list of the DEGs uniquely altered by each of the 6 PAHs is in Appendix A. To understand the functional consequences of exposure to each of these PAHs, we conducted pathway analysis on each of the PAH’s DEGs using g:Profiler [84]. The DEGs corresponded to a number of biological pathways for each PAH: retene—27, BjF—26, BkF—25, DB(a,i)P—14, DB(a,h)P—31, BbF—15. A full list of pathways enriched by each of 16 PAHs is in Appendix A.

All six Cluster B PAHs had DEG enrichment for pathways associated with cellular responses to xenobiotics, which contained the largest number of differentially expressed transcripts for retene, BkF, BjF, and DB(a,i)P. BbF and DB(a,h)P had similar DEG enrichment for cellular response to xenobiotics, but slightly more DEG enrichment for ion transmembrane transport pathways. Genes associated with response to xenobiotics common to all six PAHs were *ugt1ab, cyp1a,* and *sult6b1. Ugt1ab* is part of the family of glucuronosyl transferase enzymes involved in phase II metabolism of xenobiotics, and reported to be significantly induced when zebrafish are exposed to xenobiotic chemicals [85,86,87]. *Sult6b1* (sulfotransferase family, cytosolic, 6b, member 1) belongs to the sulfotransferase family of enzymes, which are induced in zebrafish exposed to xenobiotic chemicals, and associated with phase II metabolism of xenobiotics in mice [88,89]. Our results are concordant with previous studies that associated PAH exposures with expression changes in phase I and II metabolism genes [34,47,49]. Retene, BjF, and BkF had DEG enrichment for oxidative stress and oculomotor nerve palsy pathways. DB(a,i)P was associated with oxidative stress pathways, while DB(a,h)P had DEG enrichment for pathways associated with biological regulation, cell communication, ion transport, cellular signaling, and focal motor seizures. 

A common mechanism of action that PAH and PAH-containing mixtures have been associated with is cellular oxidative stress response, which can cause damage to both DNA and proteins [47,49,90]. Activation of the Ahr has also been linked to oxidative stress responses [91]. Our results suggest that oxidative stress response pathways may also contribute to the toxicity of retene, BkF, BjF, and DB(a,i)P. DB(a,h)P and BbF were associated with ion transport pathways which may play a causal role in their associated aberrant behavior phenotypes. Previous studies in zebrafish have reported misregulation of ion transport pathways in response to exposures to particulate matter (PM) 2.5 and arsenic [92,93]. One study found a correlation between differential expression of genes associated with calcium channels and altered behavioral development when zebrafish were exposed to the water-soluble fraction of crude oil or lead [94]. Numerous chemicals have been shown to alter the behavior of larval zebrafish; however, there is still limited understanding of the molecular mechanisms underlying altered behavioral responses [95]. The ligand dependent DEGs and pathways associated with BbF and DB(a,h)P may further our mechanistic understanding of zebrafish behavior in response to chemicals. 

### 2.8. Cluster B PAHs had Seventeen Common DEGs 

DEGs common to two or more of the Cluster B PAHs were identified. Seventeen DEGs were common to all six Cluster B PAHs (Figure 5, Appendix A), five of which were among the top 10 most elevated DEGs for Cluster B (Figure 6A; black fill, yellow text). C*yp1c1* had the highest log_2_FC for five of the six PAHs; for Db(a,i)P it was the third most differentially expressed gene. C*yp1c1* expression was highly increased by all six Cluster B PAHs and thus could be superior to *cyp1a* as biomarker of Ahr2 activation. There is not a known toxicological role of *wfikkn1 (*WAP, follistatin/kazal, immunoglobin, kunitz and netrin domain containing 1); however, its expression has been shown to significantly increase upon exposure to other PAHs and the flame retardant, mITP [49,50,66]. Future studies will be conducted to characterize the toxicological role of *wfikkn1. Sult6b1*, as mentioned previously is part of a family of Phase II detoxification enzymes. Interestingly, among the top 10 most reduced DEGs, only the uncharacterized transcript *si:dkey-65b12.6* was common to all six PAHs (Figure 6B), and *Cyp2aa11* was common to five of the six PAHs. *Cyp2aa11* is part of the *cyp2* gene family, the largest Cyp enzyme family in zebrafish [96] and, to our knowledge, its differential expression has not been associated with metabolism of any studied chemical. 

## 3. Methods

### 3.1. Chemicals

Analytical grade standards were obtained from AccuStandard (New Haven, CT, USA), Chiron Chemicals (Hawthorn, Australia), and Santa Cruz Biotechnology (SCB) (Dallas, TX, USA). Standards for the 16 PAHs were analytically verified by the Anderson Lab at Oregon State University before being dissolved in 100% DMSO to make the chemical stock solutions (Table 2). 

#### 3.1.1. Zebrafish Husbandry

Zebrafish (*Danio rerio*) of the tropical 5D line were maintained at the Sinnhuber Aquatic Research Laboratory (SARL), at Oregon State University (Corvallis, OR, USA) according to Institutional Animal Care and Use Committee protocols (ACUP 5113, date: 11th October, 2018). Adult fish were raised in densities of ~500 fish/50-gallon tank at 28 °C under a 14-hour:10-hour light:dark cycle in recirculating filtered water supplemented with Instant Ocean salts. Adult fish were fed GEMMA Micro 300 or 500 (Skretting, Inc., Fontaine Les Vervins, France) twice a day. Larval and juvenile fish were fed GEMMA Micro 75 and 150 respectively thrice a day [97]. Spawning funnels were placed in tanks the night prior, and the following morning, embryos were collected, staged, and maintained in an incubator at 28 °C in embryo media (EM) [98]. EM consisted of 15 mM NaCl, 0.5 mM KCl, 1 mM MgSO_4_, 0.15 mM KH_2_PO_4_, 0.05 mM Na_2_HPO_4_, and 0.7 mM NaHCO_3_ [99].

#### 3.1.2. Exposures

The chorions of 4 h post fertilization (hpf) zebrafish embryos were enzymatically removed using a custom automated dechorionator [100]. At 6 hpf, the embryos were placed into round bottom 96-well exposure plates (Falcon^®^, product number: 353227) with one embryo per well prefilled with 100 µL embryo medium (EM), using automated embryo placement systems [100]. Chemical stocks in 100% dry dimethyl sulfoxide (DMSO) (Table 2) were dispensed using a Hewlett Packard D300e chemical dispenser into the exposure plates [101]. Final DMSO concentrations were normalized to 1% (volume/volume) and gently shaken by the chemical dispenser during dispensing. The plates were sealed with Parafilm placed between the lid and plate to minimize evaporation, wrapped in foil to prevent exposure to light, and shaken overnight at 235 revolutions per minute (rpm) on an orbital shaker at 28°C to enhance uniform exposure [101]. Zebrafish embryos during this period of development can adapt to the dark and develop normally [102]. Embryos were statically exposed in a 28°C incubator without shaking for the remaining duration of the exposure (until 48 hpf for RNA collection, 72 hpf for immunohistochemistry, and 120 hpf for morphological analysis).

### 3.2. RNA Sequencing (RNA-seq)

#### 3.2.1. Exposure Concentrations

The data for the morphological and behavioral effects, and Cyp1a localization patterns at 120 hpf of 123 PAHs is reported in [45]. Based on the effects ranging from acutely toxic to biologically inactive in the zebrafish model, we grouped the PAHs into eight bins and selected 16 representative PAHs for our transcriptomic study. RNA sequencing test concentrations were determined by individual chemical responses. The goal was to identify a concentration for each PAH that caused an 80% phenotypic effect (EC_80_) by 120 hpf. An initial screening phase was conducted with two concentration lists depending on the stock concentration of either 10 or 1 mM based on the solubility of the PAH in DMSO. Exposure concentrations for the 10 mM stock solutions were: 50, 35.6, 11.2, 5, and 1 µM. Exposure concentrations for the 1 mM stock solutions were 5, 3.56, 1.12, 0.5, and 0.1 µM. Ten of the 16 representative PAHs did not elicit morphological effects, and transcriptomics was conducted at the highest concentration tested in the initial screening phase (50 or 5 µM). Six of the 16 PAHs induced morphological effects (see Developmental Toxicity Assessments below). For these PAHs, a well-defined concentration response was established. The lower end of the concentration range was the highest concentration that did not cause any morphological effect, and the upper end was the lowest concentration that resulted in nearly 100% mortality in the Developmental Toxicity Assessments. We fit the mean percentage of affected zebrafish for any morphological endpoint using a Hill Model (specifically a four-parameter log-logistic function). All curves were fit with the drm function for fitting dose-response models from the drc package in R [103]. This function uses least squares estimation to fit the curves. The Hill model was applied to estimate a concentration that caused 80% effects (EC_80_). The computed EC_80_ was confirmed prior to conducting the RNA sequencing. The concentrations tested for the six chemicals are listed in Table 4. The computed EC_80_ concentrations are listed in Table 2 (RNA-seq Conc.).

#### 3.2.2. Developmental Toxicity (Morphology) Assessments

At 24 hpf, embryos were assessed for 4 developmental toxicity endpoints: 24 hpf mortality, developmental progression, spontaneous movement, and notochord distortion. At 120 hpf, embryos were assessed for 18 morphology endpoints: mortality, body axis, eye, snout, jaw, otic vesicle, notochord, heart (pericardial edema), brain, somite, pectoral fins, caudal fin, yolk sac, trunk, circulation, pigment, swim bladder, and tactile response. Responses were recorded as a binary presence or absence of an abnormal morphology for each endpoint. Lowest effect levels (LELs) were calculated for each endpoint using a binomial test to estimate significance thresholds (*p* < 0.05), as previously described [101].

#### 3.2.3. RNA Isolation and Sequencing

Total RNA was isolated from pooled groups of eight zebrafish at 48 hpf with four replicates per PAH treatment group using the Direct-zol RNA MiniPrep kit (Zymo, Irvine, CA). Briefly, embryos were homogenized in 500 µL RNAzol RT (Molecular Research Center, Cincinnati, OH) with 0.5 mm zirconium oxide beads using a bullet blender (Next Advance, Averill Park, NY) for 3 min at speed 8. Samples were stored at -80°C until RNA isolation. The optional in-column DNase I digestion step was performed. Total RNA concentration and quality was determined on a SynergyMix microplate reader using the Gen5 Take3 module (BioTek Instruments, Inc., Winooski, VT). RNA integrity was confirmed (RIN score >8) using a Bioanalyzer 2100 (Agilent, Santa Clara, CA). Samples were placed in a 96 well PCR plate and submitted to Oregon State University’s Center for Genome Research and Biocomputing (Corvallis, OR) for library preparation and sequencing. Since the samples were collected on two separate days, there were two controls, one for each day. Each treatment group (16 PAHs and two controls) contained four biological replicates totaling 72 samples. Samples were prepared using the Robotic PolyA Enrichment Library Prep and Robotic Stranded RNA Library Prep Kits (WaferGen, Fremont, CA). Libraries were multiplexed and randomized across 6 lanes and sequenced with 100bp single-end reads using the Illumina HiSeq 3000. Raw sequence reads were quality checked prior to processing.

#### 3.2.4. Analysis of RNA-seq Data

Fastq files resulting from sequencing were trimmed by quality using Trimmomatic [104] and characterized using FastQC (http://www.bioinformatics.babraham.ac.uk/projects/fastqc/, accessed on: 25th Jan 2018). Transcript-level count estimates of sequence reads were accomplished using Salmon [105] against Danio rerio Zv9 transcripts maintained by Ensembl [106]. Experimental metrics across samples, encompassing trimming and pseudoalignment rates, were aggregated using MultiQC [107]. Transcript level counts were summed to the gene level using tximport [108]. Raw counts for each sample were then normalized with DESeq2 [109]. Multidimensional scaling was then used to identify outliers, those samples that were not clustered with identical replicates, and remove them before analysis continued. This resulted in a single replicate (out of four) being removed for 5 PAHs (Anthracene, DB(a,h)P, 9-MA, 3-NF, and Acenapthene). At least three biological replicates were used for each PAH treatment, with most treatments having four biological replicates. Fold changes were calculated compared to control samples, and associated *p*-values (adjusted for multiple hypothesis testing using Benjamini-Hochberg correction) were determined using DESeq2. Differentially expressed genes (DEGs) were defined as those showing a fold change compared to control of at least 1.5 with an adjusted *p*-value of less than 0.05. Pathway analysis was performed on the DEGs using g:Profiler [84].

#### 3.2.5. Network Analysis 

Networks were inferred using the Context Likelihood of Relatedness (CLR) [110] function of the MINET package in R [111]. Gene expression levels for each PAH treatment, or the controls, were averaged by mean and data was input into R using the top 500 genes, ranked by CV. CLR computes the mutual information score between all PAHs which is based on how similar the zebrafish gene expression profile resulting from PAH treatment is. PAHs inducing a more similar transcriptional response will have a higher mutual information score. CLR then converts these mutual information scores for each PAH pair into Z-scores, a measure of how many standard deviations the mutual information score for a particular PAH pair is above the average of all mutual information scores. The final output of CLR is a matrix of these Z-scores. We then defined a connector in our network as any Z-score higher than one. Any pair of PAHs whose mutual information score was at least one standard deviation above the mean was linked in the network. We also included information about the value of the Z-score by altering the thickness of the line indicating a connector, with thicker lines indicating a higher Z-score. 

### 3.3. Immunohistochemistry

#### 3.3.1. Morpholino Injections

Embryos were injected at the single-cell stage with a previously published translation-blocking morpholino (MO) targeting Ahr2 (Ahr2-MO; 5′TGTACCGATACCCGCCGACATGGTT3′) and a standard nonsense negative control (Co-MO; 5′CCTCTTACCTCAGTTACAATTTATA3′) obtained from GeneTools, Philomath, OR [80,81,112]. Approximately 2 nL of a 1.2 mM solution of Ahr2-MO or 0.95 mM Co-MO was microinjected into the yolks of tropical 5D zebrafish embryos.

#### 3.3.2. Exposures and Immunohistochemistry (IHC)

Immunohistochemistry of cytochrome P450, family 1, subfamily A (Cyp1a) protein localization was performed as previously described [113]. Based on gene expression clustering data, we conducted IHC on the PAHs we hypothesized to activate Ahr2: retene, BkF, BjF, DB(a,i)P, DB(a,i)P, and BbF. Briefly, embryos were injected with either Ahr2-MO or Co-MO, then dechorionated, and exposed from 6 to 72 hpf to the highest soluble concentration of each PAH that did not cause significant mortality. 1% DMSO was the vehicle control. At 72 hpf, ten embryos per treatment group (and two biological replicates) were euthanized with tricaine, and fixed overnight in 4% paraformaldehyde (PFA) at 4 °C. Fixed embryos were permeabilized for 10 min on ice in 0.005% trypsin, rinsed with PBS + Tween 20 (PBST) and post-fixed in 4% PFA for 10 min. Embryos were blocked with 10% normal goat serum (NGS) in PBS + 0.5% Triton X-100 (PBSTx) for 1 h at room temperature, and incubated overnight in the primary antibody mouse α fish CYP1A monoclonal antibody (Biosense Laboratories, Bergen, Norway) (1:500) in 1% NGS. An acetylated tubulin antibody (1:2000) in 1% NGS was used as a positive control for the process. Embryos were washed in PBST and incubated for 2 h in secondary antibody (Fluor 594 goat anti-mouse IgG). Eight to ten embryos per treatment group per biological replicate were assessed by epi-fluorescence microscopy using a Keyence BZ-x700 microscope at 10× magnification and scored for the presence or absence of fluorescence in the vasculature, liver, skin, neuromasts, and the yolk sac. Samples were mounted on glass slides in 3% methylcellulose, and images were acquired using the Texas Red Filter (Emission wavelength: 604–644 nm, Excitation wavelength: 542–582 nm). Exposure time was set for the Co-MO samples of each PAH and the same exposure time was used for the Ahr2-MO samples as follows: retene—1/25 s, BkF—1/50 s, BjF—1/35 s, DB(a,i)P—1/20 s, DB(a,h)P—1/30 s, BbF—1/15 s. DMSO samples were analyzed for background signal at 1/15 s.

## 4. Conclusions

In this study, we used a combination of developmental toxicity phenotypic endpoints and genome-wide transcriptomics in 48 hpf zebrafish embryos to classify 16 PAHs and identify the downstream changes in mRNA levels. Based on transcriptomic responses, the 16 PAHs clustered into two general groups. One group consisted of PAHs that were transcriptionally similar to the control, while the other group included more developmentally toxic PAHs which not only had significantly elevated *cyp1a* gene expression levels, but also primarily activated Ahr2. Interestingly, despite activating the common receptor, these PAHs each had unique ligand-dependent downstream changes in gene expression that could be the cause of their unique developmental toxicity phenotypes. The PAHs also induced Ahr2-dependent Cyp1a protein expression in the skin, which is an effective biomarker for Ahr2 activation. We show that *cyp1a* gene expression in combination with the Cyp1a protein expression patterns is an early reliable biomarker of xenobiotic Ahr activation and downstream transcriptomic changes, but an inadequate biomarker for morphological and behavioral effects from PAHs. We suggest considering other genes including *cyp1c1* and *wfikkn1*, which may also serve as reliable biomarkers for prediction of Ahr2-dependent PAH developmental toxicity. In conclusion, we have begun to characterize and classify PAHs based on their transcriptomic and developmental toxicity phenotypic responses, which will help guide the hazard characterization of other PAHs in the future and shift the focus from EPA’s 16 priority PAHs to potentially more toxic PAHs.

## Figures and Tables

**Figure 1 ijms-20-02570-f001:**
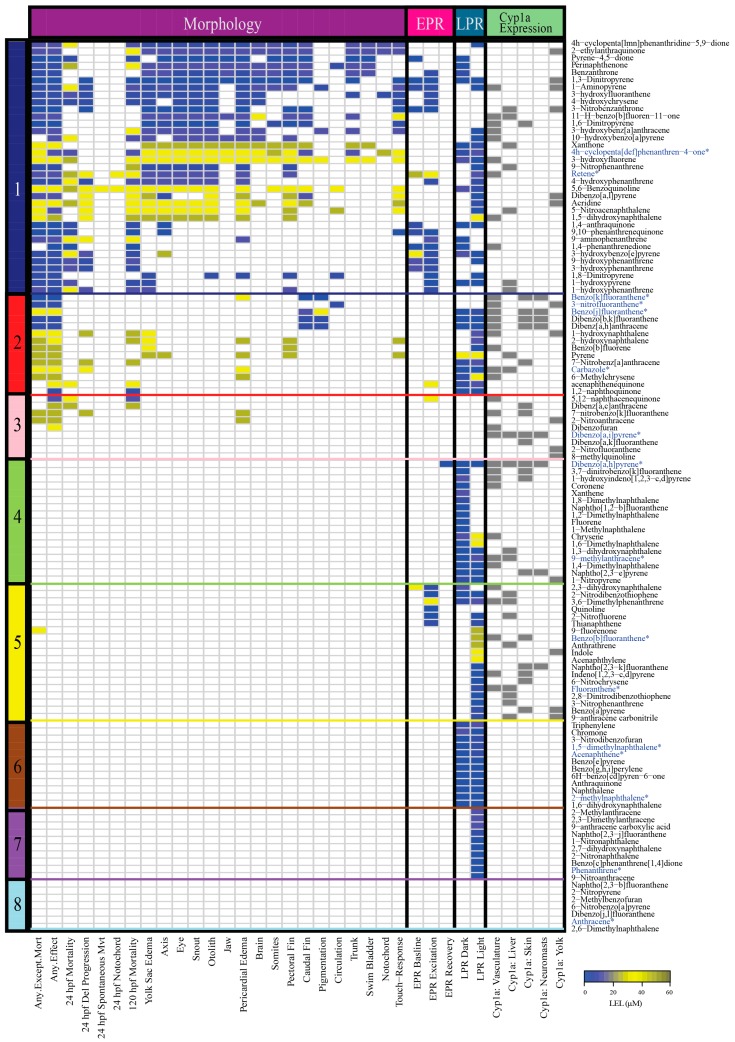
Heatmap of the morphological and behavioral responses, and Cyp1a protein localization patterns of 123 PAHs assessed using the embryonic zebrafish model. Figure illustrates the developmental toxicity (LEL) across four assays: (1) Morphology (22 endpoints) (2) Embryonic photomotor response (EPR) at 24 hpf (3) Larval photomotor response (LPR) at 120 hpf (4) Cyp1a protein expression in five tissue types. The color bar on the left side of the heatmap denotes the bins of the PAHs. The colors of the cells represent the LEL for each endpoint for each PAH. Gray denotes presence of Cyp1a expression in the specific tissue type. PAHs that were chosen from each bin for RNA sequencing analyses are denoted in blue and an asterisk. Abbreviations: Del = Delayed, Mvt = Movement, hpf = hours post fertilization.

**Figure 2 ijms-20-02570-f002:**
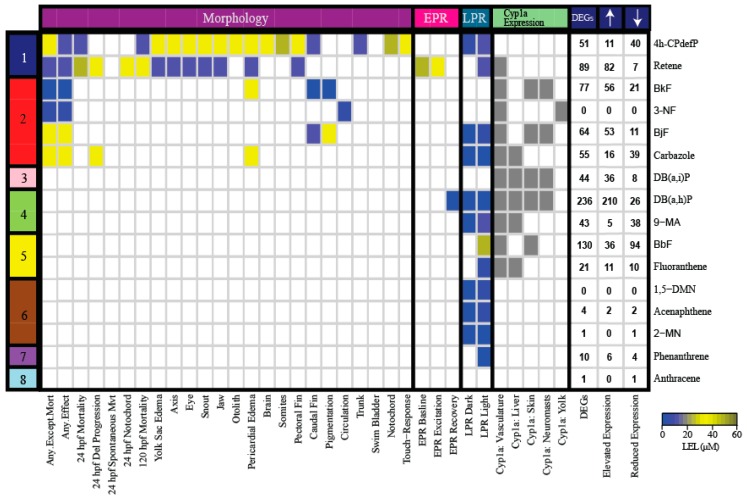
Heatmap of the morphological and behavioral endpoints, Cyp1a protein localization pattern, and DEGs associated with exposure to the 16 representative PAHs. The color bar on the left denotes the developmental toxicity bins of the PAHs selected from the 123 PAH screen. The names of the 16 PAHs are listed on the right of the heatmap. The colors in the heatmap represent the LEL concentration for each endpoint. Gray represents the presence of Cyp1a protein expression. The DEGs column represents the total number of statistically significant DEGs with FC >1.5 and PADJ <0.05. The number of DEGs with elevated and reduced expressions are also listed for each PAH. Abbreviations: Del = Delayed, Mvt = Movement, LEL = Lowest Effect Level, DEG = Differentially Expressed Gene, hpf = hours post fertilization, LPR = Larval photomotor response, EPR = Embryonic photomotor response.

**Figure 3 ijms-20-02570-f003:**
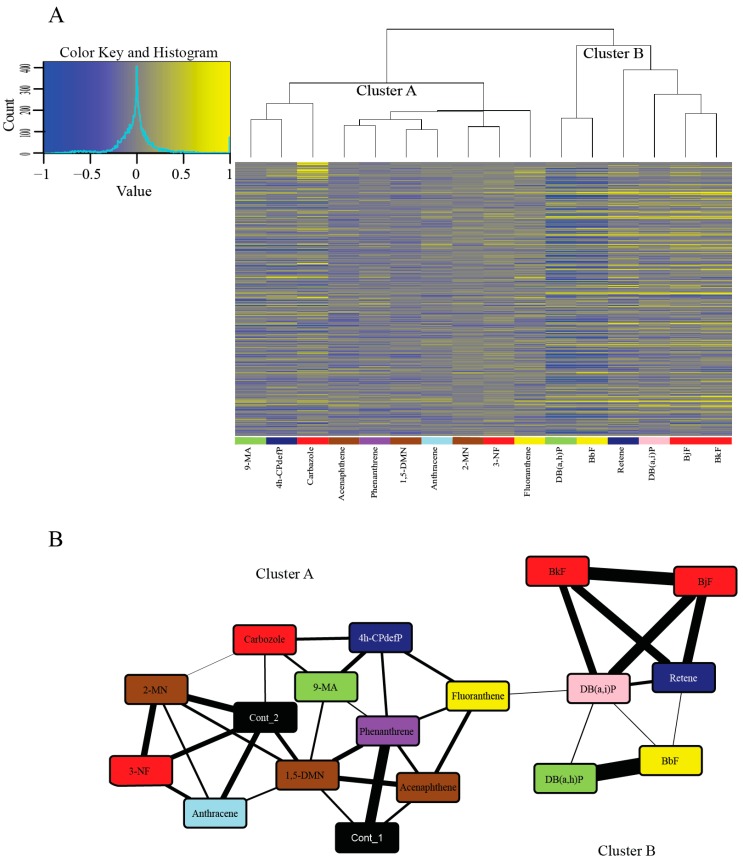
Hierarchical clustering and network analysis for the 16 PAHs. (**A**) Hierarchical clustering of the DEGs of the 16 PAHs compared to the control samples. Genes with elevated expression are in yellow and genes with reduced expression are in blue. Genes were ranked by the coefficient of variation and only the top 500 genes are included in the heatmap. (**B**) Network inferred using the Context Likelihood of Relatedness (CLR) program that links the 16 PAHs based on coordinated transcription of genes as they respond to each PAH. The nodes in the network are specific PAHs, with the colors representing the developmental toxicity bins they belong to. The two controls (in black) are included in the figure. The connectors are similarity of transcriptome response to those PAHs. The thicker the connector, the more similar the response of the PAHs. To produce a more relevant network, only the top 500 genes based on coefficient of variation (CV) was used. Abbreviations: Cont = Control.

**Figure 4 ijms-20-02570-f004:**
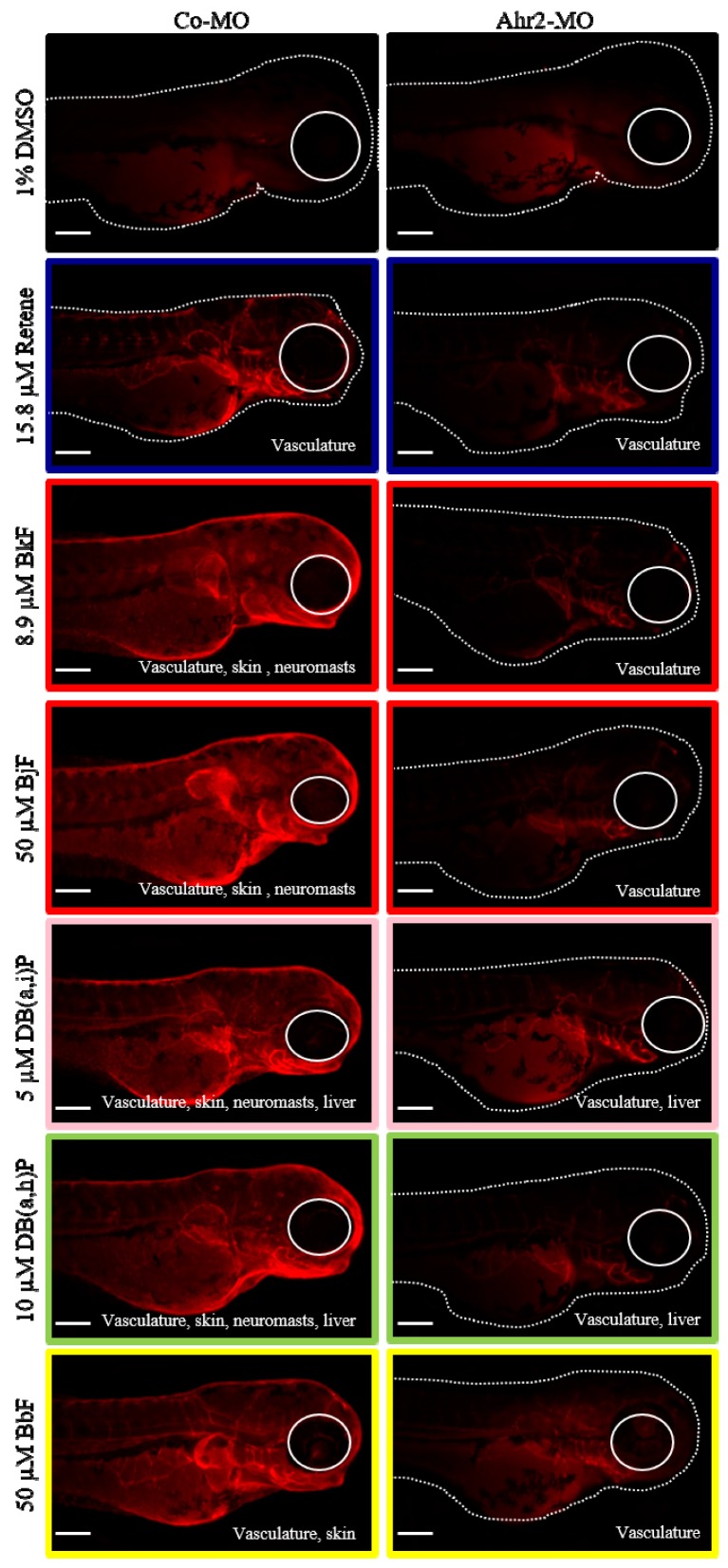
Ahr2 dependency of Cluster B PAHs. Cyp1a protein expression in 72-hpf zebrafish embryos injected with control morpholino (Co-MO) or Ahr2-morpholino (Ahr2-MO) and exposed to each of the six Cluster B PAHs (Retene, BkF, BjF, DB(a,i)P, DB(a,h)P, and BbF). 1% DMSO is the vehicle control. The color around each image represents the developmental toxicity bin the PAH belongs in. Cyp1a protein expression pattern is at the bottom right of each image. Scale bar = 300 µm.

**Figure 5 ijms-20-02570-f005:**
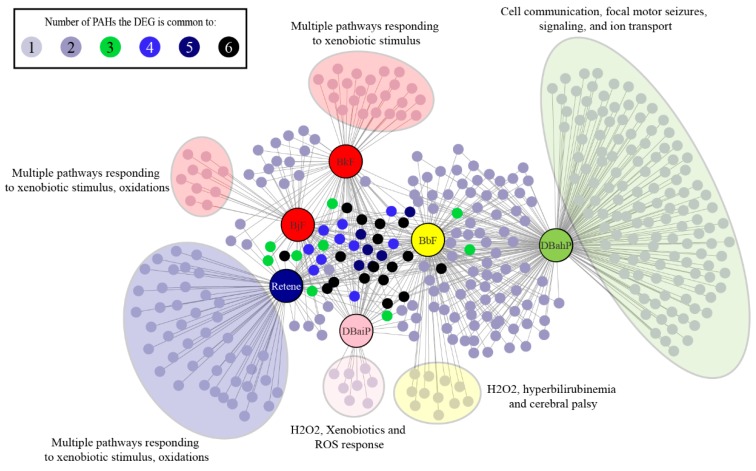
Unique DEGs (FC >1.5, adjusted *p*-value <0.05) and pathways associated with each of the six cluster B PAHs. Each cluster B PAH had DEGs associated with one to six of the other Cluster B PAHs. Each PAH also had several unique DEGs represented within the large colored clouds. The figure highlights the most significant pathways associated with each PAH.

**Figure 6 ijms-20-02570-f006:**
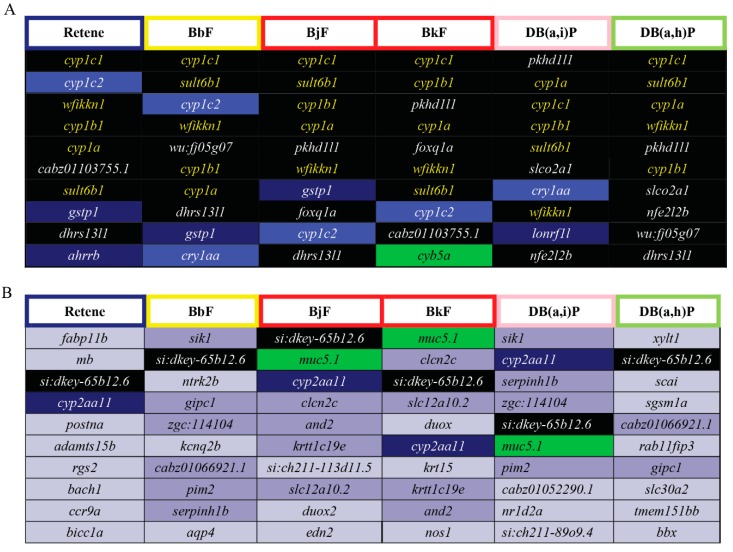
Top 10 elevated and reduced DEGs for each of the Cluster B PAHs: Retene, BbF, BjF, BkF, DB(a,h)P, and DB(a,i)P. The border colors of the cells with the PAH abbreviations represent the developmental toxicity bins they belong to. The background colors of the cells with transcript names represent the number of PAHs each differentially expressed transcript is common to amongst the 6 PAHs: Black = 6 PAHs, dark blue = 5, blue = 4, green = 3, purple = 2, gray = 1. (**A**) Top 10 elevated DEGs for each of the six Cluster B PAHs. The transcripts in yellow text are common to all six PAHs and are within the top 10 elevated DEGs. The transcripts in white text are common to all six PAHs but are not within the top 10 elevated DEGS for all six PAHs. (**B**) Top 10 reduced DEGs for each of the six Cluster B PAHs.

**Table 1 ijms-20-02570-t001:** Summary table of the eight bins the 123 PAHs were classified into with the general characteristics of each bin. The x represents the presence of morphology and behavior malformations, and *Cyp1a* protein expression.

Bin #	Endpoints		
	Morphology	Behavior	Cyp1a
1	x	x	x
2	x	x	x
3	x		x
4		x	x
5		x	x
6		x	
7		x	
8			

**Table 2 ijms-20-02570-t002:** PAHs used in this study with associated registry and use parameters. Abbreviations: Conc. = concentration.

PAH with Abbreviation	Structure	CAS	Supplier	Purity (%)	Nominal Stock Conc. (mM)	RNA-seq Conc. (µM)	IHC Conc. (µM)
4H-cyclopenta[*def*]phenanthren-4-one (4h-CPdefP)	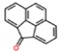	5737-13-3	Chiron	99.4	10	16.2	0.4
Retene	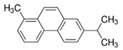	483-65-8	SCB	98	10	12.2	15.8
Benzo[*k*]fluoranthene (BkF)	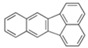	207-08-9	AccuStandard	100	10	1.9	8.9
3-nitrofluoranthene (3-NF)	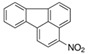	892-21-7	AccuStandard	100	10	1.9	50
Benzo[*j*]fluoranthene (BjF)	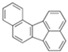	205-82-3	AccuStandard	98.1	10	14.9	50
Carbazole	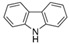	86-74-8	AccuStandard	99.7	10	50	35.6
Dibenzo[*a,i*]pyrene (DB(a,i)P)	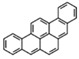	189-55-9	AccuStandard	99.8	1	5	5
Dibenzo[*a,h*]pyrene (DB(a,h)P)	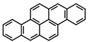	189-64-0	AccuStandard	99.9	10	10	10
9-methylanthracene (9-MA)	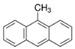	779-02-2	AccuStandard	100	10	50	50
Benzo[*b*]fluoranthene (BbF)	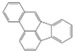	205-99-2	AccuStandard	99.2	10	50	50
Fluoranthene	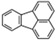	206-44-0	AccuStandard	97.2	10	50	50
1,5-dimethylnaphthalene (1,5-DMN)	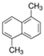	571-61-9	AccuStandard	100	10	50	50
Acenaphthene	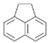	83-32-9	AccuStandard	100	10	50	50
2-methylnaphthalene (2-MN)	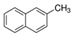	91-57-6	AccuStandard	100	10	50	50
Phenanthrene	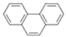	65996-93-2	AccuStandard	99.5	10	50	50
Anthracene	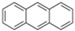	120-12-7	AccuStandard	99.6	10	50	50

**Table 3 ijms-20-02570-t003:** *Cyp1a* transcript levels and adjusted *p*-values (PADJ) associated with each of the 16 PAHs.

Bin	PAH	Cyp1a log_2_FC	PADJ
**Cluster A**			
1	4h-CPdefP	0.09	0.76
2	3-NF	0.24	0.82
	Carbazole	0.15	0.62
4	9-MA	0.31	0.13
5	Fluoranthene	0.26	0.34
6	1,5-DMN	0.04	0.76
	Acenaphthene	0.16	0.67
	2-MN	0.02	0.99
7	Phenanthrene	0.11	0.86
8	Anthracene	0.33	0.99
**Cluster B**			
1	Retene	2.06	3.05E-57
2	BkF	2.18	2.00E-64
	BjF	2.08	1.44E-58
3	DB(a,i)P	1.37	1.84E-24
4	DB(a,h)P	1.22	1.09E-19
5	BbF	1.16	2.71E-17

**Table 4 ijms-20-02570-t004:** Six PAHs that induced morphological effects with the concentrations tested to compute the EC_80._

PAH Abbreviation	Concentrations (µM) Tested to Compute the EC_80_
4h-CPdefP	8.5, 11.2, 13.9, 16.7, 19.4, 22.1, 24.8, 27.5, 30.2, 32.9, 35.6
Retene	2, 4, 6, 8, 10, 12, 14, 16, 18, 20, 22
BkF	0.5, 1.5, 2.5, 3.5, 4.5, 5.5, 6.5, 7.5, 8.5, 9.5, 10.5
3-NF	0.25, 1, 2.1, 3.3, 4.4, 5.5, 6.6, 7.8, 8.9, 10.1, 11.2
BjF	4.5, 6.9, 9.3, 11.7, 14.1, 16.5, 18.8, 21.2, 23.6, 26, 28.4
Carbazole	8.5, 12.7, 16.8, 21, 25.1, 29.3, 33.4, 37.6, 41.7, 45.9, 50

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
