# Peer review of "Coupling Genome-wide Transcriptomics and Developmental Toxicity Profiles in Zebrafish to Characterize Polycyclic Aromatic Hydrocarbon (PAH) Hazard"

_ijms, 2019, doi:10.3390/ijms20102570_

Round 1

Reviewer 1 Report

The manuscript by Shankar et al. used genome-wide transcriptomics in combination with developmental phenotype and specific protein expression to characterize polycyclic aromatic hydrocarbon (PAH) responses using zebrafish as a model. Overall, this is a powerful approach to characterize toxicological responses to classes of compounds. In practice, this is a difficult process that will require studies like the submitted manuscript to produce a standard approach. The current study is useful. However, the study illustrates the need for more studies that will identify critical developmental stages for analysis. The current study does not appear to hit the critical period. Despite this, my opinion is that this is a useful advance.

Major concerns

1. Figure 1 appears to be from another study. This was not completely obvious in the writing of the manuscript. Please make this fully explicit.

2. Conclusions about the “complex bioactivity” showing proximity of new PAHs to the PAHs examined in this study is a stretch. The granularity of the data appears to be insufficient. The authors note that additional transcriptomic and developmental phenotype studies at more developmental stages will be needed. In addition, the numbers of differentially expressed genes is low for many of the compounds tested. Finally, they note that the mixtures of PAHs that will likely exist in real-life testing situations, which will require a lot more knowledge to make predictions about new/unknown PAH samples. I think that the the writing should be clarified and conclusions should be much more conservative in this manuscript.

3. Why are there so few differentially expressed genes for the PAHs tested? Was there low coverage in the RNA-seq experiments? How many reads were made for each sample? Please clarify these details in the methods and other sections. It appears that deeper sequencing is needed in future studies.

4. Do the differentially expressed genes correlate with any of the developmental defects at 48 hpf? Even if 48 hpf is outside a critical developmental window for the genesis of the defects, the dysfunction of the organs should be revealed in the list of differentially expressed genes (e. g., heart genes in the dysfunctional heart phenotypes). This was not discussed at all in the manuscript.

5. Differences Cyp1a expression could be affected by action of the PAHs and many factors listed (metabolism, absorption, etc.). Another factor that could be discussed is the stability of compounds, which would affect the bioactivity. It appears from the methods section that the compounds were added at the beginning, but they were not replenished during the period that ended at 72 hpf.

6. The images shown in Figure 4 were not sufficient to show differential effects on liver, neuromasts, blood vessels, etc. It was clear that the Ahr2 MO reduced expression of Cyp1a, but the differential reduction in individual structures was not evident. Better labeling of the images could help, but the image quality is a bigger concern. Perhaps confocal microscopy would help.

7. Why is Ahr2 shown in all capital letters? This is not consistent with zebrafish nomenclature convention.

Minor concerns

1. Line 163: There are parentheses at the end of the line that do not contain anything [drm()]. What does “drm” refer to?

2. Line 230: the control morpholino oligonucleotide is referred to as “CO-MO.” Elsewhere, “Co-MO” is used. Be consistent.

3. Line 432: the word “range” has a space between the “n” and “g.”

Author Response

We thank the reviewers for their careful review and insightful comments. Each of the concerns has been addressed and

we believe that the manuscript is greatly improved.

Response to Reviewers:

Reviewer #1:

The manuscript by Shankar et al. used genome-wide transcriptomics in combination with developmental phenotype and specific protein expression to characterize polycyclic aromatic hydrocarbon (PAH) responses using zebrafish as a model. Overall, this is a powerful approach to characterize toxicological responses to classes of compounds. In practice, this is a difficult process that will require studies like the submitted manuscript to produce a standard approach. The current study is useful. However, the study illustrates the need for more studies that will identify critical developmental stages for analysis. The current study does not appear to hit the critical period. Despite this, my opinion is that this is a useful advance.

Thank you for your careful review of our manuscript.

Major concerns

1. Figure 1 appears to be from another study. This was not completely obvious in the writing of the manuscript. Please make this fully explicit.

This concern has been addressed by including the citation “Geier et al., 2018” in the text of the manuscript where Figure 1 is first referenced.

2. Conclusions about the “complex bioactivity” showing proximity of new PAHs to the PAHs examined in this study is a stretch. The granularity of the data appears to be insufficient. The authors note that additional transcriptomic and developmental phenotype studies at more developmental stages will be needed. In addition, the numbers of differentially expressed genes is low for many of the compounds tested. Finally, they note that the mixtures of PAHs that will likely exist in real-life testing situations, which will require a lot more knowledge to make predictions about new/unknown PAH samples. I think that the the writing should be clarified and conclusions should be much more conservative in this manuscript.

We appreciate the feedback on our conclusions. We have altered them as necessary in the text of the manuscript.

3. Why are there so few differentially expressed genes for the PAHs tested? Was there low coverage in the RNA-seq experiments? How many reads were made for each sample? Please clarify these details in the methods and other sections. It appears that deeper sequencing is needed in future studies.

This is an important question. As part of the QA/QC, each sample had very similar number of reads and exceeding 20 million, so the sequencing depth for each sample was very similar. In this study, we consistently defined differentially expressed genes (DEGs) as those showing a fold change compared to the control of at least 1.5 with and an adjusted p-value of less than 0.05. The lower number of DEGs for some of the PAHs therefore indicates that for some samples at this single developmental time point (48 hpf) very few transcripts met the criteria for differential expression.

4. Do the differentially expressed genes correlate with any of the developmental defects at 48 hpf? Even if 48 hpf is outside a critical developmental window for the genesis of the defects, the dysfunction of the organs should be revealed in the list of differentially expressed genes (e. g., heart genes in the dysfunctional heart phenotypes). This was not discussed at all in the manuscript.

The single 48 hpf time point was selected to increase the probability that we are assessing gene expression in the whole body that may be causal in developing the later phenotypes. At this time point the developmental defects are not easily visible. Correlating these early gene expression changes to individual phenotypes is a lofty goal as many of these apical endpoints (like, pericardial edema) can be caused by numerous chemical-initiated mechanisms and by targeting different cell types. By sampling all cell types simultaneously in the whole body, we believe it will be possible to use the gene expression profiling to begin to identify the specificity by which individual or groups of PH initiate toxic responses.

5. Differences Cyp1a expression could be affected by action of the PAHs and many factors listed (metabolism, absorption, etc.). Another factor that could be discussed is the stability of compounds, which would affect the bioactivity. It appears from the methods section that the compounds were added at the beginning, but they were not replenished during the period that ended at 72 hpf.

We appreciate the thoroughness of the reviewer. Differential stability of compounds in solution is definitely a possibility, and we have added this to our list of other possible factors that could affect PAH bioactivity.

6. The images shown in Figure 4 were not sufficient to show differential effects on liver, neuromasts, blood vessels, etc. It was clear that the Ahr2 MO reduced expression of Cyp1a, but the differential reduction in individual structures was not evident. Better labeling of the images could help, but the image quality is a bigger concern. Perhaps confocal microscopy would help.

The Ahr2-MO images are now labeled better. We hope this will clarify the Cyp1a expression patterns in these images – the images are dull because we kept the same exposure time settings between the Co-MO and Ahr2-MO samples of each PAH to identify reduction in expression. Confocal microscopy is a good suggestion and we will consider utilizing it for our immunohistochemistry figures in future work.

7. Why is Ahr2 shown in all capital letters? This is not consistent with zebrafish nomenclature convention.

Thank you for pointing out the error. The change has been made.

Minor concerns

1. Line 163: There are parentheses at the end of the line that do not contain anything [drm()]. What does “drm” refer to?

This is now clarified in the methods.

2. Line 230: the control morpholino oligonucleotide is referred to as “CO-MO.” Elsewhere, “Co-MO” is used. Be consistent.

We have changed each “CO-MO” to “Co-MO.”

3. Line 432: the word “range” has a space between the “n” and “g.”

The error has been fixed.

Reviewer 2 Report

Coupling Genome-wide Transcriptomics and Developmental Toxicity Profiles in Zebrafish to Characterize Polycyclic Aromatic Hydrocarbon (PAH) Hazard

General:

A well written manuscript covering the morphology and gene expression of zebrafish exposed to PAH.

The manuscript is self-critical of the limitations of the study but also is able extrapolate out form the data provided.

The authors identify where this data and techniques can be used in hazard identification/toxicity profiles, but it would be nice to see if they have planned any further work to look at different timepoints to expand on the data sets they have created and analysed to cover some of the self critical points and limitations of the study they have highlighted themselves.

Specific comments/suggestions:

Line 107: It states later in the manuscript that the 16 PAH cover all 8 bins but it would be nice to see that in the introduction when discussing the representative PAH’s

Line 134: Manufactor of 96 well plate and product code

Line 141: The exposure was in the dark to prevent compound degradation, do the authors have any concerns about the development of the embryos in the dark over the 5 days?

Table 1: Should the compound names be in bold and the subscript names in normal text for all compounds, i.e., Retene, Carbozole etc.

Table 1: Can the table be on one page and not spread out over two?

Table 1: DB(a,h)P is made up in a stock at 1mM but the RNA and HIS exposure is stating 10uM which is higher than stated in line 156 for the 1mM stock solutions, can the authors explain or was this PAH treated differently?

Line 159: States” For these PAHs, a well-defined concentration response was established between the highest concentration that did not cause any morphological effect and the lowest concentration that resulted in nearly 100% mortality.” Highest and lowest the wrong way round? I can understand what they are saying I think but it is not very clear

Line 220: Space

Figure 1: Would be easier to see the bin separation if there was a horizontal line between each bin? Also might meant that you wold not need Table 3 as the divisions between the bins can be easily seen.

Line 305-307: Feel that the sentence would be better suited to go in after the sentence at line 309.

Line 358-364: Would be nice if the listing of the individual bins follows the order of the PAH listing in Figure 1.

Figure 4: Scale bar?

Author Response

We thank the reviewers for their careful review and insightful comments. Each of the concerns has been addressed and we believe that the manuscript is greatly improved.

Response to Reviewers:

Reviewer #2:

Coupling Genome-wide Transcriptomics and Developmental Toxicity Profiles in Zebrafish to Characterize Polycyclic Aromatic Hydrocarbon (PAH) Hazard

General:

A well written manuscript covering the morphology and gene expression of zebrafish exposed to PAH.

The manuscript is self-critical of the limitations of the study but also is able extrapolate out form the data provided.

The authors identify where this data and techniques can be used in hazard identification/toxicity profiles, but it would be nice to see if they have planned any further work to look at different timepoints to expand on the data sets they have created and analysed to cover some of the self critical points and limitations of the study they have highlighted themselves.

Thank you for your careful review of our manuscript. We have included our planned future work in this manuscript as you suggested.

Specific comments/suggestions:

Line 107: It states later in the manuscript that the 16 PAH cover all 8 bins but it would be nice to see that in the introduction when discussing the representative PAH’s

This suggestion has been addressed.

Line 134: Manufactor of 96 well plate and product code

This information has been included in the Methods.

Line 141: The exposure was in the dark to prevent compound degradation, do the authors have any concerns about the development of the embryos in the dark over the 5 days?

During this period of development, zebrafish are able to adapt to the dark and develop normally. A literature reference is now included in the Methods section of the manuscript:

102. Kokel, D.; Dunn, T. W.; Ahrens, M. B.; Alshut, R.; Cheung, C. Y.; Saint-Amant, L.; Bruni, G.; Mateus, R.; van Ham, T. J.; Shiraki, T.; Fukada, Y.; Kojima, D.; Yeh, J. R.; Mikut, R.; von Lintig, J.; Engert, F.; Peterson, R. T., Identification of nonvisual photomotor response cells in the vertebrate hindbrain. J Neurosci 2013, 33 (9), 3834-43.

Table 1: Should the compound names be in bold and the subscript names in normal text for all compounds, i.e., Retene, Carbozole etc.

All compound names have been made bold.

Table 1: Can the table be on one page and not spread out over two?

We hope that with the inclusion of the graphic abstract and edits to other Tables, Table 1 will be on one page.

Table 1: DB(a,h)P is made up in a stock at 1mM but the RNA and HIS exposure is stating 10uM which is higher than stated in line 156 for the 1mM stock solutions, can the authors explain or was this PAH treated differently?

This was an error and has been corrected. The stock solution for DB(a,h)P was 10mM.

Line 159: States” For these PAHs, a well-defined concentration response was established between the highest concentration that did not cause any morphological effect and the lowest concentration that resulted in nearly 100% mortality.” Highest and lowest the wrong way round? I can understand what they are saying I think but it is not very clear

We have clarified this sentence in the methods.

Line 220: Space

The extra space has been deleted.

Figure 1: Would be easier to see the bin separation if there was a horizontal line between each bin? Also might meant that you wold not need Table 3 as the divisions between the bins can be easily seen.

Horizontal lines have been added to Figure 1. We would like to keep Table 3 as it gives a quick and easy summary of the data.

Line 305-307: Feel that the sentence would be better suited to go in after the sentence at line 309.

The change has been made.

Line 358-364: Would be nice if the listing of the individual bins follows the order of the PAH listing in Figure 1.

The changes have been made.

Figure 4: Scale bar?

Scale bars are now included.

Reviewer 3 Report

The authors used transcriptomics and developmental toxicity profiles in zebrafish to characterize PAH hazard. By using this systems biology approach they provided a start to further unravel PAH toxicity. The study is very well-conducted and well-written. As such, I have no substantial comments that could further improve the quality of this manuscript.

Author Response

We thank the reviewers for their careful review and insightful comments. Each of the concerns has been addressed and we believe that the manuscript is greatly improved.

Response to Reviewers:

Reviewer #3:

The authors used transcriptomics and developmental toxicity profiles in zebrafish to characterize PAH hazard. By using this systems biology approach they provided a start to further unravel PAH toxicity. The study is very well-conducted and well-written. As such, I have no substantial comments that could further improve the quality of this manuscript.

Thank you for your careful review of our manuscript.